# Versatile Low-Cost Volumetric 3D Ultrasound Imaging Using Gimbal-Assisted Distance Sensors and an Inertial Measurement Unit

**DOI:** 10.3390/s20226613

**Published:** 2020-11-19

**Authors:** Taehyung Kim, Dong-Hyun Kang, Shinyong Shim, Maesoon Im, Bo Kyoung Seo, Hyungmin Kim, Byung Chul Lee

**Affiliations:** 1Brain Science Institute, Korea Institute of Science and Technology (KIST), Seoul 02792, Korea; t18492@kist.re.kr (T.K.); syshim@kist.re.kr (S.S.); maesoon.im@kist.re.kr (M.I.); 2Micro Nano Fab Center, Korea Institute of Science and Technology (KIST), Seoul 02792, Korea; dhkang@kist.re.kr; 3Department of Radiology, Korea University Ansan Hospital, Korea University College of Medicine, Ansan 15355, Korea; seoboky@korea.ac.kr; 4Bionics Research Center, Korea Institute of Science and Technology (KIST), Seoul 02792, Korea; hk@kist.re.kr

**Keywords:** volumetric ultrasound imaging, freehand, low-cost, distance sensor, gimbal, inertial measurement unit

## Abstract

This study aims at creating low-cost, three-dimensional (3D), freehand ultrasound image reconstructions from commercial two-dimensional (2D) probes. The low-cost system that can be attached to a commercial 2D ultrasound probe consists of commercial ultrasonic distance sensors, a gimbal, and an inertial measurement unit (IMU). To calibrate irregular movements of the probe during scanning, relative position data were collected from the ultrasonic sensors that were attached to a gimbal. The directional information was provided from the IMU. All the data and 2D ultrasound images were combined using a personal computer to reconstruct 3D ultrasound image. The relative position error of the proposed system was less than 0.5%. The overall shape of the cystic mass in the breast phantom was similar to those from 2D and sections of 3D ultrasound images. Additionally, the pressure and deformations of lesions could be obtained and compensated by contacting the probe to the surface of the soft tissue using the acquired position data. The proposed method did not require any initial marks or receivers for the reconstruction of a 3D ultrasound image using a 2D ultrasound probe. Even though our system is less than $500, a valuable volumetric ultrasound image could be provided to the users.

## 1. Introduction

Ultrasound imaging is an extensively used technique in medical science because of its low risk of ionizing radiation and low cost compared with other medical imaging techniques, such as magnetic resonance imaging (MRI), computed tomography (CT), and positron emission tomography (PET) [1,2,3,4,5]. Conventional two-dimensional (2D) ultrasound displays the regions-of-interest (ROIs) in real-time but lacks anatomical and orientation information [4,5]. Thus, clinicians must mentally visualize a volumetric image reconstructed based on 2D ultrasound images that ultimately depends on their experience and knowledge [6,7]. Conversely, three-dimensional (3D) ultrasound provides volumetric images to visualize and depict the target objects and ROIs inside the human body [5]. Compared with 2D ultrasound, 3D ultrasound is less dependent on the operator, can achieve orientation-independent visualization, and can provide a measurement of quantitative attributes in 3D [8]. Thus far, many methods and systems have been designed and implemented for the construction of 3D ultrasound images.

The simplest way to obtain a 3D ultrasound image is to use a 2D array transducer. The elements of the 2D array transducer generate a diverging beam in a pyramidal shape, and the received echoes are processed to integrate 3D ultrasound images in real time [9,10]. Thus, 2D array transducers acquire 3D information via electronic scanning. Although 2D array transducers are capable of realizing 3D visualizations of dynamic structures in real time directly, the electrical connection of each element in 2D array transducers is much higher than that in one-dimensional (1D) array transducers. Therefore, the electrical connection or systemic load of 2D array elements is challenging [11]. In addition, a half-wavelength distance is required for the neighboring elements to eliminate the grating lobes in the sonographies [4]. In 2D array transducers, a large number of elements and minimal sizes for all elements are required that causes the manufacturing cost to rise.

Another way to obtain a 3D ultrasound image is via the use of a mechanical 3D probe in which a conventional linear array transducer is motored to rotate, tilt, or translate within the probe subject to computer control [12]. Mechanical 3D probes are relatively larger than conventional linear probes, but operate conveniently for obtaining 3D ultrasound images [4]. However, clinicians need to keep mechanical 3D probes static while acquiring images, but this can lead to potential errors when collecting data. Moreover, a particular mechanical motor is required to integrate within the transducer that lacks universality.

Conversely, a freehand scanner is flexible and convenient to operate. Clinicians can use freehand scanners to scan ROIs in any direction and position, thus allowing clinicians to select the optimal view and accommodate complex anatomical surfaces. To reconstruct a 3D image using a freehand scanner, the location and orientation of the 2D B-scan are required. Some approaches used to obtain the position data of the linear ultrasound probe have been reported. In the acoustic positioning method, three acoustic emitters are mounted on the transducer, and the microphone array is placed over the patient [13]. In this method, the microphone should be placed close to the patient to achieve an excellent signal-to-noise ratio (SNR) response. Accordingly, there should be no obstacles in the space between the emitter and microphone. A freehand ultrasound probe with an optical position system that is stable and has high accuracy consists of passive or active targets fixed on the transducer, and two or more cameras are used to track targets [14]. However, when the marker is obscured by the user’s motion, there is a problem in that the accuracy decreases rapidly owing to the lack of tracking data. Obtaining position data with a magnetic field sensor that is relatively small and flexible without the need for unobstructed sight is appropriate freehand, 3D ultrasound image reconstruction [15,16]. However, electromagnetic interference and the presence of metal objects can degrade tracking accuracy and cause distortion. Obtaining a 3D ultrasound image with a freehand linear probe requires expensive additional devices and controlled special environments to obtain accurate position data in real time.

Recently, a method for obtaining orientation or position by constructing a low-cost system have been proposed. These systems use an inertial measurement unit (IMU) and a custom fixture that limits the axis of the rotating probe [17,18]. The IMU mounted on the 2D ultrasound probe provides data on the orientation of the ultrasound image within the range of the rotation axis of the fixture while sweeping the target surface. However, to reduce positional error when scanning the surface of soft tissue, the operator will apply pressure during the scan so that the clinician can observe the deformed target. Then this can cause discomfort to the patient. Moreover, it is difficult to form an accurate volume for irregularly contoured surfaces over a wide range because the rotational range is at the point at which the acoustic coupling in contact with the skin surface is maintained. This indicates that the system can only be used in limited areas of the body.

In this study, we propose a simple method to obtain a high-quality 3D ultrasound image. To achieve this, we constructed a low-cost system that can be attached to a commercial 2D ultrasound probe. The system consisted of commercial distance sensors, a gimbal, and an IMU. The ultrasonic sensors for distance measurement are inexpensive and easy to handle, and were used to collect relevant position data. To calibrate irregular or sudden movements of the ultrasound probe with the use of the bare hand, an electronic gimbal was used to compensate for shaking or tilting. For the direction information of the ultrasound probe, an IMU was used that consisted of a three-axis accelerometer, gyroscope, and a magnetometer. The magnetometer compensates for the accumulation of drifting errors of accelerometers and gyroscopes. The IMU provides the Euler angle or quaternion values based on a comprehensive calculation and filtering of these three elements. Given that the probe allows the acquisition of position information, the proposed system can generate a high-quality 3D volumetric image that is composed of 2D ultrasound image stacks obtained from commercial 2D ultrasound probes and systems.

The proposed method does not require any initial marks or a receiver for obtaining the position data for the reconstruction of 3D ultrasound images. Moreover, use of the acquired position data of the ultrasound probe, the pressure and deformation of lesions can be obtained and compensated by contacting the probe to the skin. This study presents a simple method for 3D ultrasound imaging using a commercial 2D ultrasound probe and a very cheap system that consists of commercial components. In the following sections, the system configuration and data processing for the reconstruction of a 3D ultrasound image conducted experiments for the evaluation of the proposed system, and generated results are presented and discussed.

## 2. Materials and Methods

### 2.1. System Configuration

To obtain a high-quality 3D ultrasound image using a commercial probe, we organized a system that consisted of low-cost ultrasonic sensors, a gimbal, and an IMU, as shown in Figure 1. The system offers exact position data during the movement of the ultrasound probe by the clinician’s bare hand.

The ultrasonic sensors (HC-SR04, Shenzen AV, Shenzhen, China) used in our low-cost system were very cheap and easy to handle commercial distance sensors. The trigger signals of the ultrasonic sensors are transistor-transistor logic (TTL) pulses at a working frequency of 40 kHz and with periods of 10 µs. The measurement range of the sensor was 2–400 cm within a range of detection angles of ≤15°. The ultrasonic sensors of the system generated the sound wave that was emitted from the ultrasonic sensor that was reflected by the object’s surface. By using this time, the distance (*d*) of the sensor to the object surface was calculated as
(1)$$d\;=\;\frac{c\times t}{2}$$
where *c* is the speed of sound in air (340 m/s) and *t* is the traveling time of the ultrasound [12]. To obtain the volumetric position data, the ultrasonic sensors were held perpendicular to each other along the axes of the Cartesian coordinate (XYZ axes) in 3D space, as shown in Figure 1a. For accurate position data, measuring the change in distance of the sensor to the orthogonal surface is essential when each ultrasonic sensor is moved following the ultrasound probe. Thus, we constructed a fixed flat orthogonal space to measure the exact distance without disturbance. Each ultrasonic sensor was connected to Arduino Uno (Arduino, Italy), a microcontroller that was responsible for sending electrical signals to the ultrasonic sensor, and for receiving echo-converted electrical signals. The Arduino was connected to a personal computer by a USB.

Given that the ultrasonic probe is operated manually by clinicians, it could be accompanied by undesired and sudden dynamic movements. Owing to these unexpected motions, the directions of the three ultrasonic sensors attached to the ultrasound probe were changed, and the vertical angles between the sensors and the flat surfaces were distorted. These distortions were caused a mismatch between the real and measured distances. However, proper position data collection by the probe made it very difficult. To eliminate the distortion, it is crucial that each ultrasonic sensor is always perpendicular to the flat surface, and for this purpose, a commercial 3-axis gimbal was mounted on the ultrasound probe. The gimbal was WG2X (Feiyutech, Guilin, China) because of its light weight (238 g), shape in consideration for its combination with the ultrasound probe, load weight (130 g) of the mounted object, and availability of lock mode, a function that maintains the set direction. As shown in Figure 1a, the ultrasonic sensors were mounted to the gimbal head that was linked to a commercial probe that compensated for the error of the position data by the angular motion of the probe.

To acquire the orientation of the ultrasound probe, a commercial IMU module (EBIMUI_24GV, E2box, Hanam-si, Korea) was used that incorporated a 3-axis gyroscope, accelerometer, and a magnetometer. The IMU contained digital low-pass and Kalman filters, and transmitted the direction information to a personal computer (PC) via a serial communication link. In our low-cost 3D ultrasound system, the IMU consisted of wireless communications between the transmitter and receiver. The transmitter was equipped with an acrylic structure for mounting on the probe, as shown in Figure 1b. The receiver was connected to a PC to obtain information generated by the transmitter. By attaching the IMU to the ultrasound probe, it was possible to obtain real-time data on changes in the orientation of the ultrasound probe when the user swept the probe on the patient or the phantom. As previously reported by Herickhoff et al., the IMU, which was attached to the 2D ultrasound probe, provided the orientation data of the probe [17]. Thus, the volumetric image could be reconstructed by 2D ultrasound images using the proposed low-cost, 3D ultrasound system.

In brief, a gimbal that contained three perpendicularly combined ultrasonic sensors and an IMU were mounted on the commercial ultrasound probe for low-cost volumetric 3D ultrasound imaging, as shown in Figure 1c. In a space, such as a room in which at least three faces are orthogonal to each other, the movement of the probe in the 3D space can be represented as a rectangular coordinate by the distance data measured by the ultrasonic sensors between the wall and the probe, as shown in Figure 2a. When the user tilts the probe, as shown in Figure 2b, the gimbal compensates for the direction of the ultrasonic sensors for accurate position data in the 3D space, and the IMU provides orientation data of the ultrasound probe. With these proposed devices, 3D image reconstruction was conducted at a low cost.

### 2.2. Data Processing

To reconstruct 3D images from the proposed low-cost devices that were attached to a commercial ultrasound probe, 2D ultrasound images from the probe and location and orientation data from the proposed low-cost devices required the visualization of 3D images using operations such as mixing, cropping, volume forming, and others. For this purpose, we used two main software packages and the C++ language. We then mixed the transformed data in a 2D scanned image of the probe, and performed a 3D reconstruction of the target. Figure 3 shows a simplified block diagram of the volume reconstruction process.

The open-source software PLUS toolkit used for data acquisition for navigation and image-guides relays the communication between the tracking hardware and the imaging system so that it can mix image, orientation, and position data [19]. The orientation and position data are treated as transformed data. In addition, the PLUS toolkit presents an integrated dataset of tracked ultrasound images and includes the ability to perform 3D reconstructions with sequence data recorded with images and their transformations.

3D Slicer is an open-source software platform for medical image informatics, image processing, and 3D visualization [20]. The 3D Slicer displays volumetric images of 2D datasets and presents anatomical cross-sectional images of humans in various orientations, such as sagittal, coronal, or transverse. In addition, the transformed data received from PLUS can be visualized in the form of image data in real time.

#### 2.2.1. Transformed Data

Because PLUS and 3D Slicer process the tracking data in the form of a homogeneous transformation matrix, the physical data measured by the proposed low-cost 3D ultrasound system is converted to conform to the relevant specifications. The transform matrix *K* [21] is a 4 × 4 array expressed as
(2)$$K\;=\;\left[\begin{array}{cc} \begin{array}{cc} r_{11} & r_{12}\\ r_{21} & r_{22} \end{array} & \begin{array}{cc} r_{13} & t_{1}\\ r_{23} & t_{2} \end{array}\\ \begin{array}{cc} r_{31} & r_{32}\\ 0 & 0 \end{array} & \begin{array}{cc} r_{33} & t_{3}\\ 0 & 1 \end{array} \end{array}\right]$$
where *t*_1_, *t*_2_, and *t*_3_ represent the values of the position information related to the parallel movement in translation. Each of them is responsible for the position values along the X-, Y-, and Z-axes, and is defined by the user in 3D space [21]. The remaining *r*_11_ to *r*_33_ provides information on the tilt and rotation. The transform matrix is multiplied by a 4 × 4 array so that information about the tracking data is updated.

In our low-cost 3D ultrasound imaging, the IMU was set to a 60 Hz sample rate and orientation information was sent to the PC in the raw data. Even though the raw data can be output as a quaternion or Euler angle, the Euler angle can cause the gimbal lock. Thus, a quaternion was used to indicate the probe orientation. The general equation of the quaternion *Q* is given by Equation (3) [22].
(3)Q = w+ix+jy+kz
where *w* is the real part, and *ix*, *jy*, and *kz*, are imaginary parts. In addition, *i*, *j*, and *k*, are three orthogonal unit vectors in the spatial vector in Equation (3) that satisfy *i*^2^ = *j*^2^ = *k*^2^ = *ijk* = −1 [22]. *Q* can be converted into a 3 × 3 matrix as follows,
(4)$$M\;=\;\left[\begin{array}{c} 1-2y^{2}\;-2z^{2}\\ 2xy-2wz\\ 2xz+2wy \end{array}\;\begin{array}{c} 2xy+2wz\\ 1-2x^{2}-2z^{2}\\ \;2yz-2wx\; \end{array}\;\begin{array}{c} 2xz-2wy\\ 2yz+2wx\\ 1-2x^{2}-2y^{2} \end{array}\right]$$

*M* corresponds to *r*_11_ to *r*_33_ of Equation (2). By obtaining matrix data, the part related to the rotation of the transform data is represented by IMU data. Position data corresponding to *t*_1_, *t*_2_, and *t*_3_ in Equation (2) were measured by three ultrasonic sensors that were connected to the Arduino module. The outputs of the ultrasonic sensors connected to the Arduino had a sample rate of 23 Hz. A digital low-pass filter (cutoff frequency: 10 Hz) was applied to the Arduino module, and the measured data units were calculated in millimeters. The PC converted the measurement information of the distance between a fixed object and a moving sensor into parallel movement sensor data. The parallel movement of the low-cost 3D ultrasound system was updated in real-time by continually applying the difference in distances measured previously and currently. When the transform data is represented in the 3D slicer, the position data corresponding to *t*_1_, *t*_2_, and *t*_3_ in Equation (2), represented the LR (left, right), PA (posterior, anterior), and IS (inferior, superior), respectively. These correspond to the three virtual axes in Figure 2a. The data from three ultrasonic sensors were arbitrarily designated by the operator and assigned to *t*_1_, *t*_2_, and *t*_3_, according to the desired direction of the movement. As a result, the transform data were obtained by the rotation information of the IMU and the parallel movement information of the three ultrasonic sensors.

In short, the tracking data obtained from the hardware was transformed into 4 × 4 matrix data and communicated with PLUS. This process was programmed in C++ using Visual Studio Express (Microsoft, Redmond, WA, USA) with the OpenIGTLink (Department of Radiology, Brigham and Women’s Hospital, Boston, MA, USA) library, an open-source network communication interface designed for image-guided interventions [23,24].

#### 2.2.2. Volumetric Imaging

The ultrasound probe used to acquire the image was VF13-5, which is a linear array transducer with a 13–15 MHz bandwidth and a 60 mm maximum display depth. Ultrasound images (30 frames per second) were transmitted from the ultrasound system (ACUSON Antares Ultrasound System, SIEMENS, Munich, Germany) to the PC (Intel i5-7600 processor and 32 GB of random access memory) through a capture device (USB Capture HDMI Gen 2, MAGEWELL, Nanjing, China). The 2D scan image of the ultrasound system appeared on the display, and provided various information or parameters. Given that only the anatomical section of the ultrasound image was needed, it was cropped to fit the area of interest.

The tracked images combined with the ultrasound images and the transformed data were sent from PLUS to the 3D Slicer to observe the movement in real time. The scanning data around the area of interest were recorded to create a meta-image file that contained the sequence, and a 3D volume was then formed using the file via PLUS.

### 2.3. Experimental Setup

To evaluate the proposed 3D ultrasound imaging method, the accuracy of the position data measured by the ultrasonic sensor had to be estimated in advance. The XYZ-stage table used the stepping motors (BM-56L, FasTech, Bucheon-si, Korea), and was used to move precisely the low-cost 3D ultrasound system. Figure 4 shows the XYZ stage and the low-cost 3D ultrasound system setup for the measurement of the position error. The low-cost 3D ultrasound system was mounted on the XYZ stage holder to control precisely the movement in each direction. The flat planes for the reflection of the ultrasound by the ultrasonic sensors were positioned at distances of 400, 300, and 500 mm along the X, Y, and Z axes, respectively. The XYZ stage was moved at various distances (10, 20, 30, 40, and 50 mm) in each axial direction, and the accuracy of the measured position data was evaluated by comparing the actual movement of the XYZ stage with the distance measured by the ultrasonic sensors. In addition, the change in error of the measured position data in the repeated motion (five cycles) was evaluated.

An ultrasonic needle breast biopsy phantom (CIRS, New York, NY, USA) was used to evaluate the 3D image reconstruction with the proposed low-cost system. The ROI in the phantom was discovered by prescanning with an ultrasonic probe. Figure 5a shows the ultrasound image of the cystic mass at the breast biopsy phantom. To verify the proposed low-cost 3D ultrasound imaging, an ultrasound probe with commercial devices, such as the gimbal, ultrasonic sensors, and IMU, was swept several times along the contours of the breast phantom surface with a bare hand to obtain position data, as shown in Figure 5b. During this process, a set of image data with position information was recorded and reconstructed by PLUS to create a 3D volumetric ultrasound image.

During the ultrasound scans, changes in morphology of the soft tissue by the ultrasound probes that pressed the tissue surfaces caused changes in the lesion. To identify these changes, changes in the soft tissue surface according to the applied pressure were measured from the ultrasound images and position data.

## 3. Results and Discussion

### 3.1. Evaluation of Position Data

In 3D ultrasound imaging using low-cost commercial devices, such as a gimbal, ultrasonic sensors, and an IMU, the accuracy of the position data is very important to reconstruct a volumetric image. To evaluate the position data obtained from the ultrasonic sensors attached to the gimbal, the XYZ stage was used to move the commercial ultrasound probe precisely. The actual movement of the probe by the XYZ stage and the measured position data from the ultrasonic sensors were compared, and the position error was then computed.

Figure 6 shows the actual movement along the X-, Y-, and Z-axes (the direction of each axis is shown in Figure 4) and the measured position data. The motor of the XYZ stage was controlled by a PC to move the ultrasound probe at a distance of 100 mm from the starting point. The velocity of the ultrasound probe that moved by the XYZ stage was set to 4.81 mm/s. The XYZ stage was set to move in the direction of each axis after 60 s to confirm that the ultrasonic distance sensors immediately responded to the movement of the ultrasound probe moving by the XYZ stage. In Figure 6, the black line depicts the actual movement of the XYZ stage, and the blue line depicts the measured distance of the probe by the ultrasonic sensors. As shown in Figure 6a–c, the measured distance by the sensors increased immediately when the actual distance of the XYZ stage increased. This is because the gimbal compensated for the irregular motion caused by the movement of the probe. Consequently, the ultrasonic sensors attached to the gimbal provided the data of the distance moved by the probe without any delay. The error between the actual distance by the XYZ stage and the distance measured by the ultrasonic sensors was determined. The mean absolute errors were 0.79, 1.25, and 1.09 mm in the X, Y, and Z-axes, respectively. Given that the error of the commercial ultrasonic sensors of the proposed system is related to the measurement distance, it is important to evaluate the error rate as the relative error. The relative errors between the actual and measured distances for each axis were 0.212% (*X*-axis), 0.323% (*Y*-axis), and 0.262% (*Z*-axis), as shown in Figure 6d. The distance error of each axis by the single movement of the probe was deficient (lower than 0.4%). This means that the proposed method using devices attached to a commercial ultrasound probe can provide the position data successfully.

When moving the ultrasound probe with the user’s bare hand to observe the ROIs, the ultrasonic probe underwent repeated irregular movements. To obtain more accurate position data from the actual movement of the probe by hand, it was confirmed that the error of the distance data measured by the ultrasonic sensor could be kept low despite repeated movements of the ultrasonic probe. The ultrasound probe moved by 100 mm in each axial direction using the XYZ stage and then returned to the initial position. After each movement, the ultrasound probe was held stationary in 30 s to analyze the distance data errors. To determine the distance data error in a repetitive motion, it was repeated five times. Figure 7 shows the measured distance data during successive repetitions of the movement. The relative errors of the movement with respect to the X-, Y-, and Z-axes were 0.245%, 0.361%, and 0.405%, respectively. During the five cycle movements, the relative error did not increase or decrease. It has been proven that accurate position data can be obtained by the proposed devices attached to the commercial probe by maintaining the error at very low levels, even in repetitive motions.

In summary, according to the movement of the commercial probe with the three perpendicularly placed ultrasonic sensors on the gimbal, the position data are associated with relatively low errors. Thus, the proposed method can provide very accurate position data for 3D ultrasound image reconstructions by constructing very inexpensive commercial devices.

### 3.2. Volumetric Image Reconstruction

It was confirmed that the position data of the ultrasound probe, which was obtained via the proposed low-cost commercial devices, such as a gimbal, ultrasonic sensors, and an IMU, were relatively accurate. A commercial ultrasound probe, including the proposed low-cost devices, was controlled with a bare hand to scan the breast phantom, and 2D ultrasound images, location data, and direction information were acquired every 1/30 s. Using the obtained position and direction data, 2D ultrasound images (1247 frames) were stacked to reconstruct a 3D volumetric ultrasound image stack. As shown in Figure 8a, the 2D ultrasound images were positioned along the contour of the breast phantom. Owing to the irregular pressure developed while scanning the phantom with a bare hand, the breast phantom was pressed irregularly. Therefore, the images were arranged unevenly. Given that the lesion of the breast phantom was located inside the reconstructed 3D ultrasound image, the lesion was verified by sectioning the image in the ultrasound scan direction (XZ plane), which is orthogonal to the directions of the 2D ultrasound images (YZ plane), as shown in Figure 8b. For the analysis of the lesion shape, the 3D ultrasound image was cropped around the cystic mass of the breast phantom, and the cross-section of the lesion was verified in the XZ (Figure 8c) and YZ planes (Figure 8d). The reconstructed volumetric ultrasound image that was cropped and centered on the cystic mass successfully determined the shape of the lesion of the breast phantom in 3D. The proposed inexpensive system was attached to the commercial ultrasound probe (VFX13-15, Siemens Healthineers, Erlangen, Germany). The lateral resolution of a single frame of ultrasound image was identified using the phantom (ATS-539 multi-purpose ultrasound phantom) and was 0.136 mm. As scanning the breast phantom with the scanned length of 70 mm, the 1247 frames of 2D ultrasound images were reconstructed as a 3D ultrasound image. The resolution of the 3D reconstruction was 0.056 mm in theory. However, the position data acquired with the three ultrasonic distance sensors did not provide an accurate ultrasound probe position due to the sensors’ measurement error. Therefore, the reconstructed 3D ultrasound image was slightly blurred, as shown in Figure 8c,d. If the accuracy of the position data is enhanced by using a more accurate distance sensor, the higher resolution of the 3D ultrasound image can be obtained, but the system’s price increases due to the accurate sensor cost.

To evaluate the accuracy of the reconstructed 3D ultrasound image using the proposed low-cost method, it was analyzed in comparison with the 2D ultrasound image. Figure 8e is a conventional 2D ultrasound image scanned with a 2D ultrasound probe in the XZ plane. Figure 8f shows the cross-section of the XZ plane at the location of the lesion based on the reconstruction of the 2D ultrasound image scanned in the YZ plane in 3D with the proposed method. The shapes of the cystic mass were similar in the 2D ultrasound image and the corresponding cross-section of the 3D ultrasound image. In the 2D ultrasound image, shown in Figure 8e, the most extended length of the cystic mass in the breast phantom was 18.4 mm. At the cross-section of the 3D ultrasound image by the proposed method, as shown in Figure 8f, the most extended length of the cystic mass was 19.6 mm. When the 2D ultrasound image and the cross-section of the 3D ultrasound image were compared, the difference in the most extended length of the cystic mass was 1.4 mm. This was attributed to the error of the position data obtained from the ultrasonic sensors owing to the uneven pressure applied to the smooth phantom surface while the ultrasound probe was used to scan with a bare hand.

Using the proposed cheap system (the total cost was less than $500), a 3D ultrasound image was reconstructed with 2D ultrasound images with a commercial probe. The proposed system reconstructed a 3D ultrasound image stack by storing 2D ultrasound images at 30 frames per second while scanning a lesion with a bare hand. Therefore, if the speed of sweeping the ultrasonic probe is high, the number of 2D ultrasound images in the same volume decreases, and the sizes of the voxel for the reconstruction of the 3D ultrasound image also increase. As the size of the voxel increases, the spatial resolution of the 3D image decreases, while the imaging speed increases owing to the decreased amount of data that need to be processed. The proposed system reconstructed the 3D ultrasound image in 50 s by combining the 2D ultrasound images of 1247 frames. Morgan et al. reconstructed the abdominal phantom into a 3D ultrasound image using 730, 2D ultrasound image frames with a reconstruction time of 36.5 s per frame [18]. Our proposed system combines 2D ultrasound image frames in a relatively fast reconstruction time, so it is possible to obtain more precise 3D ultrasound images.

Our inexpensive system can be easily adapted to 2D imaging probes and systems already in use in the clinic to reconstruct 3D ultrasound images. However, given that image reconstruction is performed using the open-source software PLUS toolkit and 3D Slicer, optimization of the image processing methodology is required. Additionally, there is a limit to the accuracy of the position data because it uses inexpensive ultrasonic distance sensors. To increase the accuracy of position data, a more precise distance sensor must be used, but this increases the price. To increase the quality of 3D ultrasound images, more accurate position data and image processing software should be used, but this is a trade-off relationship with the price of the system. Therefore, we need to optimize the method we propose in the future to find the optimum price and 3D ultrasound image quality.

### 3.3. Lesion Contraction Based on Probe Pressure

As mentioned above, when a target lesion is scanned using an ultrasonic probe with the user’s hand, the probe presses the soft tissue and a force is exerted by the operator’s hand. At this time, the shape of the lesion may also change as the soft tissue is pressed and contracted. The method proposed in this study can acquire accurate position data with an ultrasonic sensor used in all axis directions. Therefore, the pressed volume of the soft tissue can be calculated as the length by the position data. In this study, we proposed a method to verify the correlation between changes in the shape of a lesion and pressure caused by a bare hand during scanning with an ultrasound probe.

Figure 9 shows the ultrasound images of the breast phantom with the entire scan area up to the bottom part of the phantom as the pressure change owing to the probe. To obtain correct measurements, the direction of the ultrasound probe was maintained vertical to the ground. First, the cystic mass of the breast phantom was scanned with an ultrasonic probe with a minimum pressure, as shown in Figure 9a. For comparison, the cystic mass was scanned with an increased pressure. As shown in Figure 9b, the breast phantom and the cystic mass appeared to be more compressed than that shown in Figure 9a. In addition, the difference in the length from the surface of the phantom to the bottom part was measured by the ultrasound image and was found to be 6.48 mm. At this time, the difference in the position data from the ultrasonic sensor was 5.85 mm. The difference between the measured ultrasound image and position data was only 0.65 mm (9.72%).

In the clinical environment in which the ultrasound probe is used, reconstruction of the volume image stack by the 2D ultrasound image may cause errors frequently if the position data of the probe is not obtained correctly. Conversely, based on the proposed method, the error caused by the irregular pressure of a bare hand can be reduced. Moreover, if the proper mechanical properties of the soft tissue are offered, the shape change caused by irregular pressure will be compensated, and a real 3D ultrasound image of the lesion can be generated without the application of an external pressure. This method paves the way for low-cost 3D ultrasound images without the application of external pressure.

## 4. Conclusions

A simple method was proposed to obtain a high-quality 3D ultrasound image. For this purpose, we used low-cost commercial devices, such as three distance sensors, a gimbal, and an IMU, and attached them to a commercial 2D ultrasound probe. Inexpensive ultrasonic distance sensors were used to collect relevant position data. The latter were attached to a gimbal to calibrate irregular or sudden movements of the ultrasound probe with a bare hand. To obtain the direction information of the probe, an IMU was used. The distance and direction data were sent to a PC and were combined with 2D ultrasound images to reconstruct a 3D ultrasound image stack. The position data obtained by three low-cost commercial ultrasonic sensors yielded a low error <0.5%. The reconstructed 3D high-quality ultrasound image was composed of 1247 2D ultrasound images. Moreover, use of the acquired position data of the ultrasound probe, the pressure and deformation of lesions could be obtained and compensated by contacting the probe to the surface of the soft tissue.

The proposed method did not require any initial marks and receivers for obtaining position data to reconstruct the 3D ultrasound image with the use of the 2D ultrasound probe. Moreover, although the cost of our system was less than $500 (the cost of the commercial ultrasound system was excluded), a valuable volumetric ultrasound image could be provided to the users. Our method can be applied simply by attaching inexpensive commercial devices to a commercial 2D ultrasound probe. In addition to its simple configuration, it has good accessibility owing to open-source software used to control it. This method will pave the way for cost-effective, hand-held, ultrasound probe systems for obtaining 3D ultrasound images.

## Figures and Tables

**Figure 1 sensors-20-06613-f001:**
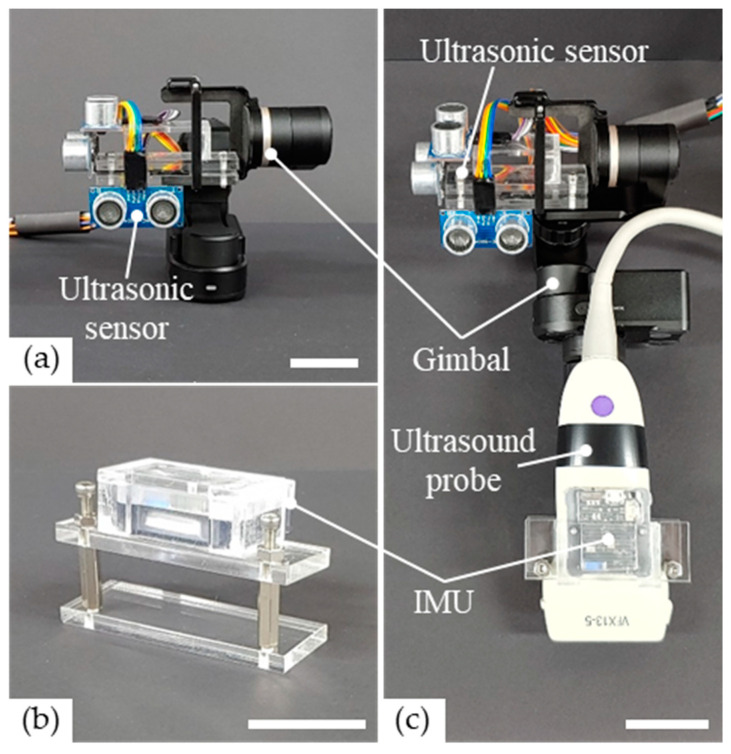
Photographs of the proposed low-cost devices attached to the commercial ultrasound probe. (**a**) Three ultrasonic sensors are placed perpendicularly to each other and are combined with a gimbal to acquire the position data of the probe. (**b**) The inertial measurement unit (IMU) and the poly(methyl methacrylate) (PMMA) housing were combined easily with the probe to obtain orientation data of the commercial ultrasound probe. (**c**) Overall, the proposed low-cost devices are attached to a commercial probe for the acquisition of volumetric ultrasound images. All scale bars are 35 mm.

**Figure 2 sensors-20-06613-f002:**
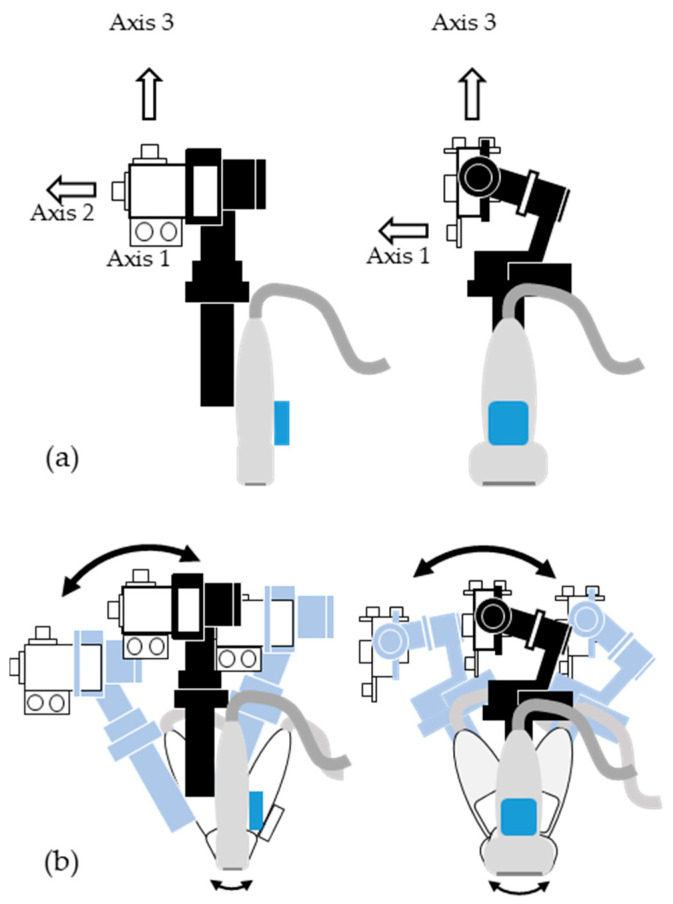
Schematics of (**a**) the axes of three ultrasonic sensors for measuring the distance to the vertical flat plane to identify the position of the probe, and (**b**) the gimbal compensating the direction of the three ultrasonic sensors at irregular angular movements.

**Figure 3 sensors-20-06613-f003:**
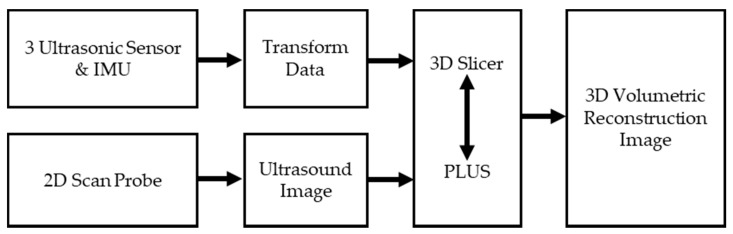
Block diagram for data acquisition and three-dimensional (3D) reconstruction of two-dimensional (2D) ultrasound images by the proposed method. The transformed data from ultrasonic sensors and the IMU, and the 2D ultrasound images from a commercial probe, were combined to reconstruct a 3D volumetric ultrasound image using 3D Slicer and PLUS. One transform data per ultrasound image was tagged that contained position and orientation of the probe.

**Figure 4 sensors-20-06613-f004:**
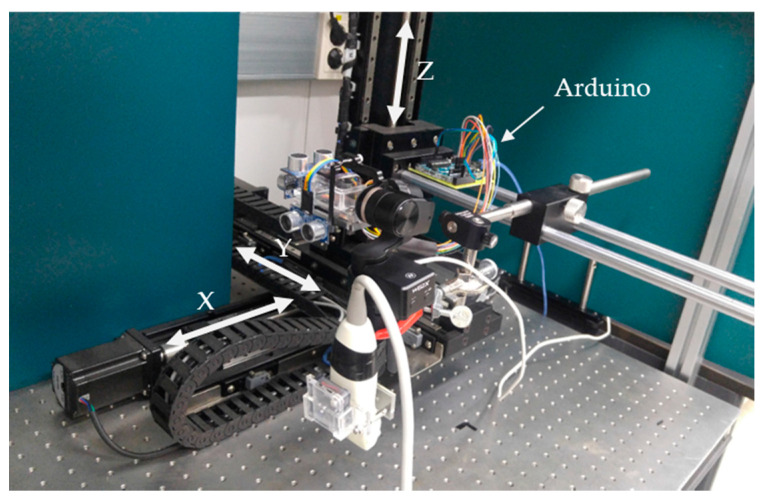
Experimental setup for the identification of the position data with the use of the XYZ stage, including the ultrasound probe with low-cost devices.

**Figure 5 sensors-20-06613-f005:**
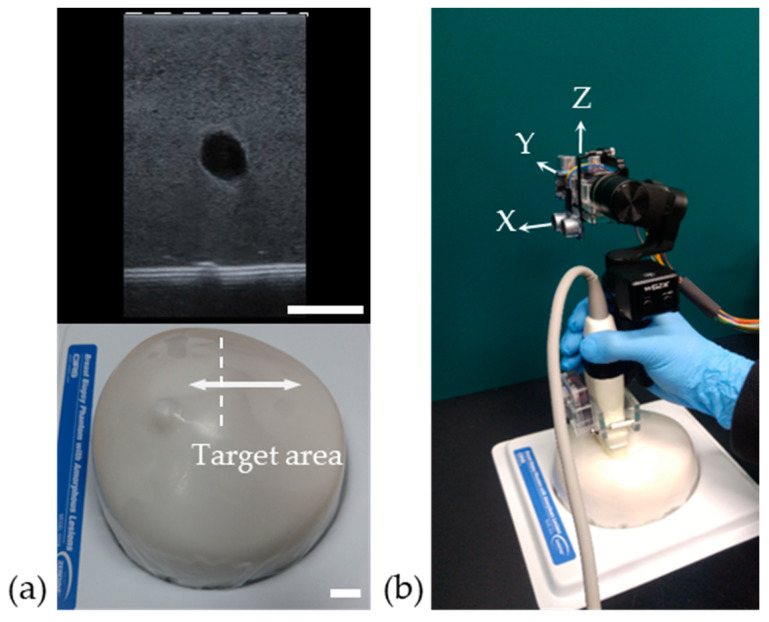
(**a**) Breast biopsy phantom containing a cystic mass, and the corresponding ultrasound image. The dashed lines of the upper and lower figures indicate the same spatial ranges. (**b**) Image of the target area (cystic mass in the breast phantom) using the ultrasound probe that contains the low-cost devices for position data acquired with a bare hand. All scale bars are 15 mm.

**Figure 6 sensors-20-06613-f006:**
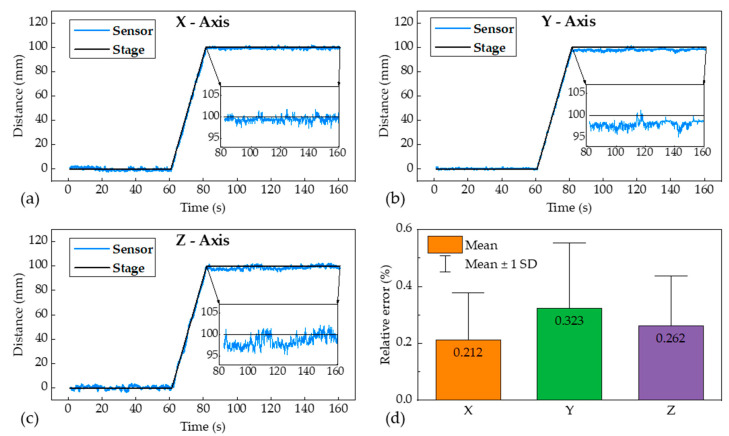
Actual movement (black line) of the probe by the XYZ stage and the measured position data (blue line) of the probe by the ultrasonic sensors along the (**a**) X-, (**b**) Y-, and (**c**) Z-axes. (**d**) The relative error of the measured position data of the ultrasonic sensors in each axis (calculated from (**a**–**c**)).

**Figure 7 sensors-20-06613-f007:**
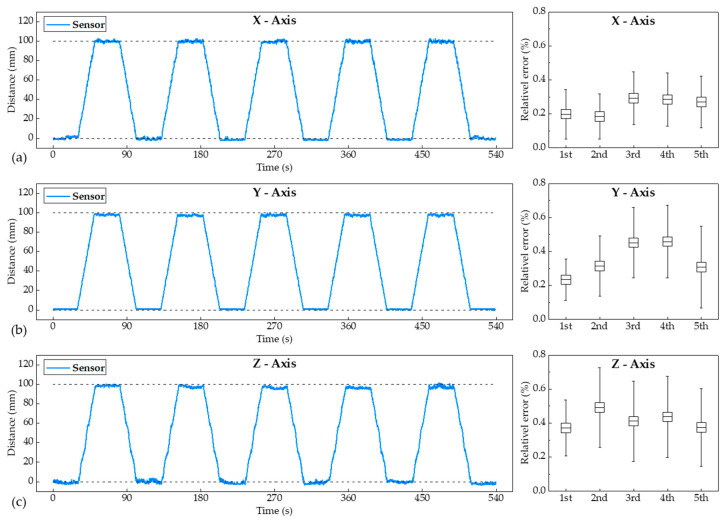
Measured position data and the relative error from the ultrasonic sensors following five cycles of repeated movements along the (**a**) X-, (**b**) Y-, and (**c**) Z-axes.

**Figure 8 sensors-20-06613-f008:**
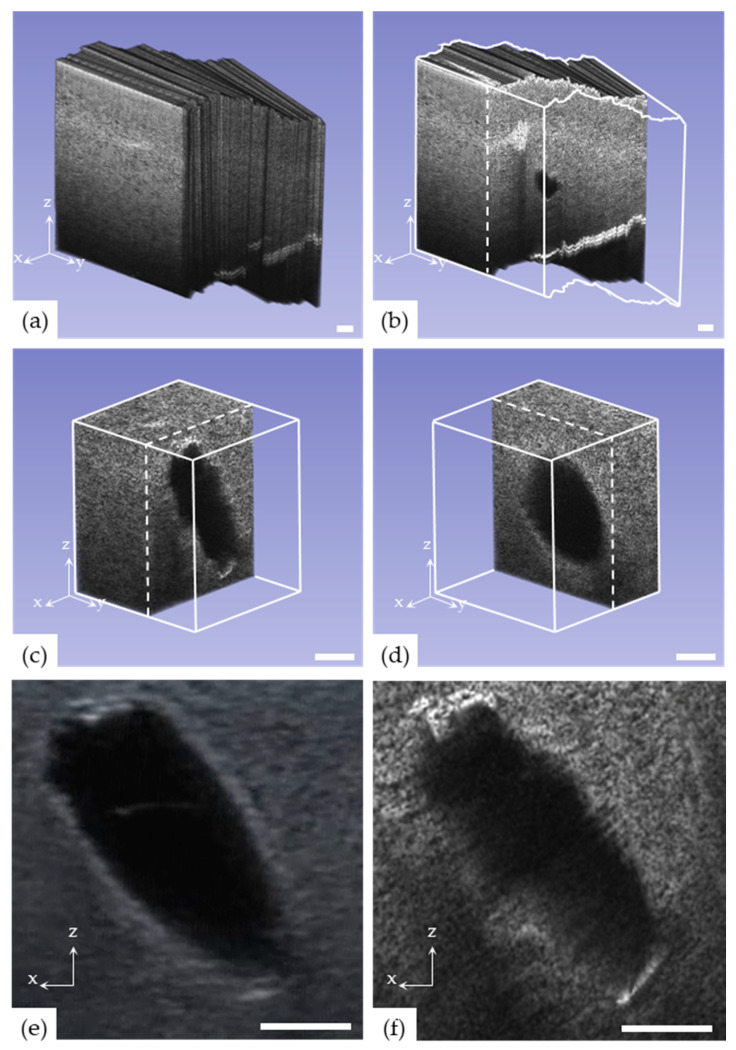
Volumetric reconstructed image from 2D ultrasound images acquired with bare hand scanning using the proposed method. (**a**) Shape of the volumetric reconstruction following the breast phantom surface. (**b**) Cross-section (perpendicular to 2D ultrasound image) for the detection of the cystic mass (the target area) in the breast phantom. For more accurate analysis, cropped volumetric images around the cystic mass were cross-sectioned in the (**c**) XZ and (**d**) YZ planes. Comparison of the target ultrasound image between (**e**) a conventional 2D ultrasound image and (**f**) the cross-section of the XZ plane from a reconstructed 3D ultrasound image.

**Figure 9 sensors-20-06613-f009:**
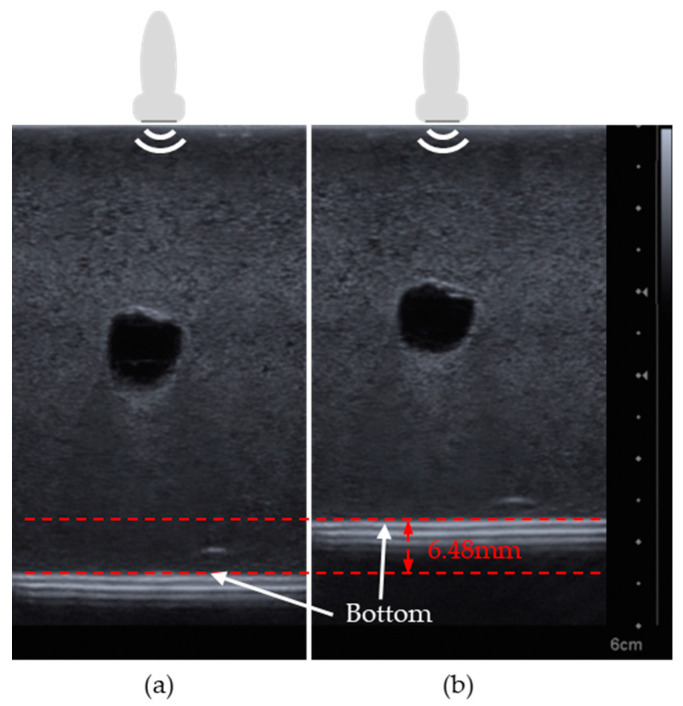
Ultrasound images of breast phantom (**a**) before and (**b**) after pressure is exerted. The scale bar is on the right side of the figure.

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
