# Peer review of "Versatile Low-Cost Volumetric 3D Ultrasound Imaging Using Gimbal-Assisted Distance Sensors and an Inertial Measurement Unit"

_sensors, 2020, doi:10.3390/s20226613_

Round 1

Reviewer 1 Report

This manuscript reported a simple method for obtaining a high-quality 3D ultrasound image, through integrating three distance sensors, a gimbal, and an IMU, and 2D ultrasound probe. The position data obtained by the ultrasonic sensors can have an error less than 0.5%. And the reconstructed 3D ultrasound image shows a high quality, composed of 1247 2D ultrasound images. But before recommending for publication, I have the following comments for the authors.

  1. Some denotation can be added in Figure 1 to clearly show the different components of the system. Dash lines in Figure 5 are a bit thin.
  2. How is the data for position measurement by ultrasound sensors in the x, y and z axis. Why is the error in y axis is larger than the other two axes (Figure 6)?
  3. How is the scanning speed of the system? Will the vibration of hands have a strong effect on the measurement results?
  4. How is the data from the IMU sensor?
  5. Can the system perform other direction of scanning other than the vertical scanning?

Reviewer 2 Report

This manuscript reports the development of a low-cost 3D ultrasound imaging system based on a 1D array ultrasound probe and a spatial tracking unit. This design is interesting and could be useful for 3D ultrasound imaging in resource-limited settings. The manuscript is also well written and easy to follow. In considering of the above, I think the manuscript has its merit. However, my major concern for this manuscript is that its technical validation is rather weak and needs to be significantly enhanced before publication. 

1) the accuracy of the 3D ultrasonic distance sensing was expressed with relative values are less meaningful compared to absolute errors, which are related to the spatial resolution of the reconstructed 3D ultrasound images. The authors are encouraged to present the absolute errors in x, y, and z directions and discuss its implications on the spatial resolution.

2) How about the accuracy of the IMU that registers the rotation and orientation of the probe.

3) Quantitative experiments are required to be performed to study the spatial resolution of the reconstructed 3D ultrasound images. 

4) Sec 3.1 consist content that should be moved to the Materials and Methods section. 

5) What's the frame rate of 3D imaging.

6) A discussion section should be included, in which the author should discuss advantages and limitations of the developed technique including the accuracy, spatial resolution, imaging speed in comparison of existing techniques in literature.

Reviewer 3 Report

Dear Authors; After reading your manuscript, I have found a new design of a low-cost system attached to a commercial 2D ultrasound probe which consists of commercial ultrasonic distance sensors, a gimbal, and an inertial measurement unit (IMU), but I could not found an explanation (good details) for the construction of 3D ultrasound image from 2D images. I list you some comments asking you read it carefully and make the required modifications, which are: 1- At the end of Introduction section you have to write the organization of your manuscript. 2- Give the Reference of Equations 1, 2, and 3. 3- Page 6 of 14, Line 225, give the reason for choosing the sample rate of 23Hz. 4- Page 6 of 14, Line 226, what is the benefit of using the Low pass filter, and what its cutoff frequency? 4- You have not explained the method of converting from 2D to 3D! 5- You have depends on what measure to consider the image a high quality image?

Round 2

Reviewer 1 Report

The authors have fully addressed my comments, and I recommend the acceptance of the paper.

Author Response

Thank you for your kind acceptance and agreements on all the comments.

Reviewer 2 Report

It is an interesting concept, however, without the evaluating the spatial resolution, the value of the proposed imaging system can not be assessed, which significantly reduced the impact and quality of this work. 
